# Cardiac Magnetic Resonance Findings in Patients Recovered from COVID-19 Pneumonia and Presenting with Persistent Cardiac Symptoms: The TRICITY-CMR Trial

**DOI:** 10.3390/biology11121848

**Published:** 2022-12-18

**Authors:** Dagmara Wojtowicz, Karolina Dorniak, Marzena Ławrynowicz, Piotr Wąż, Jadwiga Fijałkowska, Dorota Kulawiak-Gałąska, Joanna Rejszel-Baranowska, Robert Knut, Maciej Haberka, Edyta Szurowska, Marek Koziński

**Affiliations:** 1Department of Cardiology and Internal Medicine, Institute of Maritime and Tropical Medicine, Medical University of Gdańsk, 81-519 Gdynia, Poland; 2Department of Noninvasive Cardiac Diagnostics, Medical University of Gdańsk, 80-211 Gdańsk, Poland; 3Department of Nuclear Medicine, Medical University of Gdańsk, 80-210 Gdańsk, Poland; 4Second Department of Radiology, Medical University of Gdańsk, 80-214 Gdańsk, Poland; 5Department of Radiology, Medical University of Gdańsk, 80-214 Gdańsk, Poland; 6Department of Cardiology, School of Health Sciences, Medical University of Silesia, 40-055 Katowice, Poland

**Keywords:** non-ischemic cardiac injury, cardiac magnetic resonance imaging, coronavirus disease 2019, late gadolinium enhancement, myocarditis, SARS-CoV-2

## Abstract

**Simple Summary:**

The high prevalence of persistent cardiovascular symptoms in patients after coronavirus disease 2019 (COVID-19) recovery emphasizes the need for the investigation of cardiac complications. We present the results of the TRICITY-CMR study designed as a single-center, cross-sectional study. Our major goal was to estimate the prevalence of non-ischemic cardiac injury (defined as active myocarditis and/or late gadolinium enhancement (LGE)) on cardiac magnetic resonance (CMR). Additional purposes were to compare clinical characteristic and CMR findings between patients hospitalized and non-hospitalized during the acute phase of COVID-19 pneumonia. Our results indicate that at least half of patients with persistent cardiovascular symptoms present with non-ischemic cardiac injury in CMR, mainly manifested as LGE lesions. The majority of LGE lesions were located in the left ventricle at inferior and inferolateral segments at the base. Active myocarditis was found in the minority of individuals. Moreover, the need for hospitalization during the acute phase of COVID-19 was not associated with a greater risk of non-ischemic cardiac injury. Further studies are required to determine the long-term cardiovascular consequences of COVID-19.

**Abstract:**

The prevalence and clinical consequences of coronavirus disease 2019 (COVID-19)-related non-ischemic cardiac injury are under investigation. The main purpose of this study was to determine the occurrence of non-ischemic cardiac injury using cardiac magnetic resonance (CMR) imaging in patients with persistent cardiac symptoms following recovery from COVID-19 pneumonia. We conducted a single-center, cross-sectional study. Between January 2021 and May 2021, we enrolled 121 patients with a recent COVID-19 infection and persistent cardiac symptoms. Study participants were divided into those who required hospitalization during the acute phase of SARS-CoV-2 infection (n = 58; 47.9%) and those non-hospitalized (n = 63; 52.1%). Non-ischemic cardiac injury (defined as the presence of late gadolinium enhancement (LGE) lesion and/or active myocarditis in CMR) was detected in over half of post-COVID-19 patients (n = 64; 52.9%). LGE lesions were present in 63 (52.1%) and active myocarditis in 10 (8.3%) post-COVID-19 study participants. The majority of LGE lesions were located in the left ventricle at inferior and inferolateral segments at the base. There were no significant differences in the occurrence of LGE lesions (35 (60.3%) vs. 28 (44.4%); *p* = 0.117) or active myocarditis (6 (10.3%) vs. 4 (6.3%); *p* = 0.517) between hospitalized and non-hospitalized post-COVID-19 patients. However, CMR imaging revealed lower right ventricular ejection fraction (RVEF; 49.5 (44; 54) vs. 53 (50; 58) %; *p* = 0.001) and more frequent presence of reduced RVEF (60.3% vs. 33.3%; *p* = 0.005) in the former subgroup. In conclusion, more than half of our patients presenting with cardiac symptoms after a recent recovery from COVID-19 pneumonia had CMR imaging abnormalities indicating non-ischemic cardiac injury. The most common finding was LGE, while active myocarditis was detected in the minority of patients. CMR imaging abnormalities were observed both in previously hospitalized and non-hospitalized post-COVID-19 patients. Further research is needed to determine the long-term cardiovascular consequences of COVID-19 infection and the optimal management of patients with suspected post-COVID-19 non-ischemic cardiac injury.

## 1. Introduction

Since March 2020, coronavirus disease 2019 (COVID-19) has quickly spread throughout the globe. Subsequently, the pandemic has led to millions of deaths worldwide and exerted an unprecedented impact on healthcare systems, international economy, and everyday life [1]. Although the majority of COVID-19 cases are asymptomatic or mild, vulnerable individuals may develop potentially fatal severe acute respiratory syndrome-coronavirus-2 (SARS-CoV-2) infection with concomitant multiple organ dysfunction.

Apart from pulmonary manifestation, numerous cardiovascular complications (e.g., acute myocardial injury, acute heart failure, pulmonary embolism, arrhythmias, acute coronary syndrome, myocarditis, pericardial effusion) were reported in patients infected with SARS-CoV-2 [2]. Importantly, the occurrence and severity of non-ischemic cardiac injury may not be necessarily related to the severity of COVID-19 pneumonia. Possible pathomechanisms of COVID-19 cardiovascular complications may be associated with angiotensin-converting enzyme 2-related signaling pathways (widely expressed both in the lungs and in the cardiovascular system) [3], cytokine storm, respiratory failure with hypoxia, and drug interactions [4,5]. Previous studies indicated that 20–30% of hospitalized COVID-19 patients develop acute myocardial injury with elevated cardiac troponins, which is associated with unfavorable prognosis [6,7].

Interestingly, in a post-mortem pathological case study, fulminant myocarditis with heart tissue testing positive for SARS-CoV-2 RNA was observed without any evidence of pneumonia [8]. To date, several studies have reported on the occurrence of myocarditis in post-COVID-19 patients [9,10]. However, the prevalence, clinical course, detailed cardiac magnetic resonance (CMR) characteristics, and long-term consequences of COVID-19-related myocarditis are still under investigation.

CMR remains the gold standard, non-invasive imaging tool in patients with suspected myocarditis, and recommendations regarding CMR protocol in COVID-19 patients with suspected non-ischemic cardiac injury were recently published [11].

The main purpose of this study was to determine the occurrence of non-ischemic cardiac injury using CMR imaging in patients with persistent cardiac symptoms after the recovery from COVID-19 pneumonia. Additionally, we aimed to (i) characterize CMR findings in post-COVID-19 patients, and (ii) compare CMR findings between patients who required hospitalization due to COVID-19 pneumonia and those non-hospitalized.

## 2. Materials and Methods

### 2.1. Study Design and Population

The TRICITY-CMR trial was designed as a single-center, cross-sectional study. The results of its pilot study were previously published in a research letter [12]. Here we report the detailed study methodology together with the final results of the overall study population.

Consecutive adult patients fulfilling all inclusion criteria and without any exclusion criterion were recruited at the outpatient post-COVID-19 cardiology clinic (Department of Cardiology and Internal Medicine, Institute of Maritime and Tropical Medicine, Medical University of Gdańsk, Gdynia, Poland). Study candidates with persistent cardiac symptoms were referred by family doctors, cardiologists, pulmonologists, and specialists in infectious diseases from outpatient clinics as well as physicians from dedicated COVID-19 hospital wards from the Pomerania region. Information on the activity of our outpatient post-COVID-19 cardiology clinic was widely disseminated by local print, broadcasting, and internet media. Study inclusion criteria comprised (i) SARS-CoV-2 infection confirmed using a reverse transcription polymerase chain reaction (RT-PCR) swab test within 1 month before the occurrence of new cardiac symptoms, (ii) lack of clinical symptoms of ongoing COVID-19 pneumonia, and (iii) persistent symptoms indicating heart involvement (e.g., chest pain, palpitation, dyspnea, fatigue). The exclusion criteria were as follows: (i) a history of prior cardiac disease (except for arterial hypertension), (ii) explanation for patient’s complaints other than COVID-19 infection, and (iii) contraindications for CMR examination (e.g., cardiac implantable electronic devices, metallic intraocular foreign bodies, moderate, severe or end-stage chronic kidney disease (stages 3B–5 according to the Kidney Disease Improving Global Outcomes classification), history of allergic reactions to gadolinium-based contrast agent, claustrophobia). We excluded from the study patients with known cardiac disease in order to minimize confounding factors (i.e., to reduce the probability of detecting non-COVID-19-related CMR abnormalities). We considered for study participation both patients previously hospitalized and non-hospitalized for COVID-19 pneumonia. Patients were enrolled between January 2021 to May 2021.

All candidates for study participants were initially screened by an experienced cardiologist familiar with the diagnostics and therapy of COVID-19 infection. For all patients, we collected detailed medical history and performed a physical examination comprising a 12-lead electrocardiogram (ECG) together with transthoracic echocardiography. Additional procedures (i.e., 24 h ECG Holter monitoring, ambulatory blood pressure monitoring, laboratory tests) were performed when clinically indicated. Based on clinical data and available test results, extracardiac etiology of symptoms was excluded in all study candidates. The severity of the acute phase of COVID-19 infection was assessed according to the National Institute of Health guidelines [13].

Clinical and demographic data, treatment methods, laboratory test results, and CMR imaging measurements were collected and analyzed using Microsoft Excel Spreadsheet software (Microsoft Corporation, Redmond, WA, USA). The study population was divided into two subgroups: patients who required hospitalization during the acute phase of COVID-19 infection and those who were not hospitalized. Comparisons were made between these subgroups.

The study was designed and conducted in accordance with the Declaration of Helsinki, and its protocol was approved by the Independent Bioethics Committee for Scientific Research at the Medical University of Gdańsk (Approval No. NKBBN/47/2021). Written informed consent was obtained from all study participants.

### 2.2. Study Endpoints

The primary study endpoint was CMR-confirmed non-ischemic cardiac injury defined as a composite of CMR-diagnosed active myocarditis and/or the presence of non-ischemic late gadolinium enhancement (LGE). The CMR diagnosis of myocarditis was based on 2018 updated Lake Louise Criteria [14], while LGE was defined as an unequivocal area of hyperintensity with signal intensity exceeding that of the remote myocardium by at least five standard deviations. Secondary study endpoints were the components of the primary study endpoint. Additional endpoints included CMR-assessed left ventricular ejection fraction (LVEF), the presence of reduced LVEF (<57%) [15], left ventricular end-systolic volume (LVESV), left ventricular end-diastolic volume (LVEDV), left ventricular stroke volume (LVSV), myocardial mass, global longitudinal relaxation time (T1), prolonged global T1 (>1035 ms; institutional reference range of 951–1035 ms), global transverse relaxation time (T2), prolonged global T2 (>49 ms; institutional reference range of 39–49 ms), global extracellular volume (ECV) fraction based on the commonly adopted formula incorporating blood hematocrit, right ventricular end-diastolic volume (RVEDV), right ventricular ejection fraction (RVEF), reduced RVEF (<52% and <51% in men and in women, respectively) [15], the presence of pericardial effusion, and computed tomography (CT)-assessed extent of the COVID-19 pneumonia.

### 2.3. CMR Scanning Protocol and Image Analysis

All participants underwent CMR examination with 1.5 Tesla scanners (Magnetom Area or Magnetom Sola, Siemens AG, Erlangen, Germany, with an 18-element phased array cardiac coil) using standardized imaging protocols. These included long axis and short axis cine series for anatomy and functional assessment, followed by cardiac parametric mapping sequences for longitudinal (T1) and transverse (T2) relaxation time measurement (MOLLI (Modified Look-Locker) sequence) for T1 and a T2-prepared bSSFP sequence for T2, respectively (MyoMaps, Siemens Healthineers, Erlangen, Germany), as well as routine LGE assessment in the long axes and a short axis stack using both fast single-shot bSSFP inversion recovery and segmented phase-sensitive inversion recovery sequences, performed within 7–15 min after injection of 0.1 mmol/kg of gadobutrol (Gadovist, Bayer AG, Leverkusen, Germany) [16]. The left ventricle was divided into 17 segments, according to the American Heart Association (AHA) [17]. All CMR images were evaluated by the same two physicians (a cardiologist and a radiologist, both with long-standing experience in CMR) using commercial software (SyngoVia VB40, Siemens Healthineers, Erlangen, Germany). Any doubts were resolved by a consensus decision after discussion with a third experienced CMR reader. Persons analyzing CMR scans were unaware of the results of laboratory biomarkers.

### 2.4. Lung Assessment with Computed Tomography

The subgroup of patients hospitalized due to COVID-19 pneumonia underwent chest CT upon hospital admission. CT facilitates the diagnosis of COVID-19 pneumonia and enables a precise evaluation of the extent of the lung tissue affected. All non-contrast CT chest examinations were performed using a SOMATOM go.Top scanner (Siemens Healthineers, Erlangen, Germany) according to the manufacturer’s recommendations. Study participants were scanned in a supine position from the superior thoracic aperture to the lung base at the end of inspiration. The lesions characteristic for COVID-19 pneumonia on CT images are visible as ground glass opacity that may co-exist with consolidations and a crazy paving pattern. An extent of the involved lung tissue by COVID-19 pneumonia was calculated with automatic postprocessing, artificial intelligence-based, commercially available software (SyngoVia VB30A CT Pneumonia Analysis software, Siemens Healthineers, Erlangen, Germany).

### 2.5. Laboratory Measurements of Cardiac and Inflammatory Biomarkers

Peripheral venous blood samples were obtained from all post-COVID-19 patients prior to the CMR procedure. Blood samples were analyzed immediately after blood collection in a local laboratory. Well-established laboratory biomarkers of myocardial injury and necrosis (high-sensitivity cardiac troponin I (hs-cTnI)), heart failure (N-terminal pro-B-type natriuretic peptide (NT-proBNP)), and inflammation (C-reactive protein (CRP)) were measured on the Alinity i analyzer using commercially available tests (Alinity i STAT High Sensitivity Troponin-I, Alere NT-proBNP for Alinity i, Alinity c CRP Vario Reagent Kit; all assays manufactured by Abbott Laboratories, Wiesbaden, Germany). The assay-specific cutoff values indicating elevated concentrations for hs-cTnI (>0.016 ng/mL for females and >0.034 ng/mL for males), NT-pro-BNP (>100 pg/mL), and CRP (>5 mg/L) were applied.

### 2.6. Statistical Analysis

The results were generated using the R statistics language (R 4.0.5 environment, R Core Team, Vienna, Austria). Basic statistics such as median values and the first and the third quartiles were calculated for quantitative variables. The Shapiro–Wilk normality test was used to verify the hypothesis that quantitative data come from a normally distributed population. Comparing two samples from the population with a normal distribution, the homogeneity of their variance was checked. For analysis, the appropriate tests were selected (Wilcoxon rank sum test with continuity correction or Student’s *t*-test). For qualitative variables, the numbers (and percentage) of items in individual categories were calculated. The basic tests for checking the independence of the qualitative variables were the Pearson’s χ^2^ test with Yates’ continuity correction or Fisher’s exact test for count data when appropriate. The assumed significance level was α = 0.05. All *p*-values were rounded to three digits.

## 3. Results

### 3.1. Study Course and Participant Characteristics

A total of 121 post-COVID-19 patients were enrolled in the study. The demographic and clinical characteristics are presented in Table 1.

In the majority of the studied post-COVID-19 patients, the predominant complaints were dyspnea and fatigue that could not be explained by lung involvement. At the time of study enrollment, none of the included patients reported any symptoms of acute COVID-19 pneumonia. Almost one-third of the study participants had arterial hypertension, and approximately one-quarter of them were obese. Chronic respiratory diseases were present in 16 individuals, including 14 (11.6%) patients previously diagnosed with bronchial asthma and 2 (1.7%) with obstructive sleep apnea.

During acute phase of COVID-19 pneumonia, slightly more than half of all study participants experienced mild illness, while moderate, severe, and critical disease developed in 14 (11.6%), 40 (33.1%), and 3 (2.5%) of the studied patients, respectively.

Among the hospitalized post-COVID-19 study participants, the predominant reason for hospital admission was severe or critical illness (74.2% of the patients). The remaining patients were hospitalized either due to serious symptoms (e.g., fever > 40 °C) or markedly elevated inflammatory biomarkers (e.g., CRP concentration > 100 mg/L).

The median time from the diagnosis of COVID-19 infection to CMR examination was 41 (25; 61) days.

Almost half of the studied individuals required hospitalization during the acute phase of COVID-19 pneumonia. COVID-19 patients admitted to the hospital compared with non-hospitalized COVID-19 study participants had a higher risk profile (i.e., were older and more likely to be obese or to have a history of respiratory diseases). Non-contrast chest CT imaging showed typical radiological findings for COVID-19 in the lungs (e.g., bilateral crazy paving, ground-glass opacities) in all hospitalized study participants. Among them, the mean extent of high-opacity COVID-19 abnormalities according to the CT pneumonia analysis algorithm was 26.3%. From a total of 58 hospitalized patients, the majority (n = 41; 70.7%) received oxygen supplementation. Two of them required high-flow nasal cannula oxygen therapy. Additionally, 72.4% of the hospitalized study participants (42 of 58) received systemic corticosteroids, 27.6% (16 of 58) remdesivir, 5.2% (3 of 58) convalescent plasma, and 3.4% (2 of 58) tocilizumab.

Importantly, the median time from COVID-19 diagnosis to CMR imaging was markedly shorter in hospitalized vs. non-hospitalized post-COVID-19 study participants.

### 3.2. Laboratory Tests

Laboratory measurements of blood samples collected prior to CMR examination revealed that only two study participants (1.7%) presented with elevated hs-cTnI concentrations (>0.016 ng/mL for females and >0.034 ng/mL for males). Both of them were hospitalized during the acute phase of COVID-19 pneumonia. Concentrations of hs-cTnI were similar in hospitalized and non-hospitalized post-COVID-19 patients.

Additionally, 30 study participants (24.8%) had significantly elevated NT-proBNP concentration (>100 pg/mL), 18 of which required hospitalization during the acute phase (*p* for the comparison of hospitalized and non-hospitalized study participants = 0.127). There were no significant differences in the serum creatinine or hemoglobin concentrations, or in white blood count, including lymphocyte count, between hospitalized and non-hospitalized post-COVID-19 study participants. Importantly, CRP concentration was markedly higher in hospitalized vs. non-hospitalized post-COVID-19 patients. The number of patients with elevated CRP concentration (>5 mg/L) was also higher in the former group, but the difference did not reach the threshold for statistical significance (16 (27.6%) vs. 5 (7.9%); *p* = 0.051).

### 3.3. CMR Findings in Hospitalized vs. Non-Hospitalized Post-COVID-19 Patients

#### 3.3.1. Conventional and LGE Sequences

The prevalence of CMR abnormalities among the study population is shown in Table 2. Non-ischemic cardiac injury was visualized by CMR in 54.5% of the post-COVID-19 study participants, without any difference between hospitalized and non-hospitalized post-COVID-19 patients. Among 66 post-COVID-19 patients with non-ischemic cardiac injury, abnormal LGE images were seen in all except 1 individual (from the non-hospitalized group). Additionally, CMR lesions indicating ongoing myocarditis were demonstrated in 10 post-COVID-19 study participants (8.3%), including 6 (10.3%) and 4 (6.3%) patients in the hospitalized and non-hospitalized subgroup, respectively. There were no significant differences in the occurrence of LGE lesions or active myocarditis between hospitalized and non-hospitalized post-COVID-19 patients.

Most post-COVID-19 study participants with non-ischemic cardiac injury had at least two affected segments (39 of 64 patients with non-ischemic cardiac injury; 60.9%). In twelve post-COVID-19 patients (18.8%), we observed three-segment non-ischemic cardiac injury, and in eight post-COVID-19 study participants (12.5%), four-segment non-ischemic cardiac injury was reported. Notably, in one of post-COVID-19 patients, six left ventricular segments were involved. The majority of LGE lesions were located in the left ventricle at inferior and inferolateral segments at the base. Myocardial LGE distribution in the studied patients is shown in Figure 1.

Examples of COVID-19-related cardiac lesions, visualized using CMR imaging, in our study participants are shown in Figure 2.

Reduced LVEF (<57%) was observed in 38.8% (n = 47) post-COVID-19 patients. Fifty-six post-COVID-19 study participants (46.3%) had reduced RVEF (<52% for men and <51% for women). The RVEF, but not LVEF, values in hospitalized post-COVID-19 patients were significantly lower when compared with those who were not hospitalized (Table 2; Figure 3). Importantly, none of the post-COVID-19 study participants presented with LGE lesions localized within the right ventricle.

Small pericardial effusion was seen in two post-COVID-19 patients (one hospitalized and one non-hospitalized).

#### 3.3.2. Mapping Sequences

There were no significant differences in native myocardial T1 and T2 relaxation times and ECV between hospitalized and non-hospitalized post-COVID-19 patients (Table 2). Interestingly, we found a trend towards a higher rate of patients presenting with prolonged T2 relaxation time (>49 ms) in hospitalized vs. non-hospitalized post-COVID-19 patients.

Among ten post-COVID-19 patients with CMR-confirmed active myocarditis (seven males and three females, median age: 57 (range 37–78) years), nine had prolonged T1 and T2 relaxation times and LGE, and only one patient fulfilled T2-STIR (short tau inversion recovery) and LGE diagnostic criteria for ongoing myocarditis. The most common cardiac symptoms in this subgroup were persistent dyspnea (n = 5; 50%) and fatigue (n = 3; 30%) after the recovery from COVID-19 pneumonia. Half of these patients had a history of hypertension, three had diabetes, and two were obese. The median time between COVID-19 diagnosis and CMR imaging in patients with CMR features of active myocarditis was 37.5 days (range: 22–106 days). Most of these individuals had reduced LVEF (n = 7; 70%) and three post-COVID-19 patients with CMR-confirmed active myocarditis presented with reduced RVEF (30%).

### 3.4. Comparison of Post-COVID-19 Patients with and without Non-Ischemic Cardiac Injury

Post-COVID-19 patients with non-ischemic cardiac injury were older, had higher cTnI concentration, and were more likely to have hyperlipidemia and more severe COVID-19 pneumonia than those without non-ischemic cardiac injury (Table 3). However, both subgroups had a comparable extent of involved lung tissue detected via chest CT scans. Additionally, cardiac complaints after recovery from COVID-19 pneumonia were similar in patients with and without non-ischemic cardiac injury.

Upon CMR examination, post-COVID-19 patients with non-ischemic cardiac injury had significantly lower LVEF together with higher LVESV and myocardial mass than their counterparts without non-ischemic cardiac injury (Table 4). Additionally, we found trends towards longer global T1 and higher RVEDV in the former subgroup.

## 4. Discussion

### 4.1. Major Study Findings

In the present research, we report CMR findings from 121 consecutive patients presenting with cardiac symptoms after the recovery from COVID-19 pneumonia. Non-ischemic cardiac injury was confirmed by CMR in more than half of our study participants, and the most frequent abnormality was LGE lesions, present in 98.4% of the patients with evidence of non-ischemic cardiac injury. The majority of LGE lesions were located in the left ventricle at inferior and inferolateral segments at the base. Importantly, active myocarditis was infrequent in our study participants, and occurred only in 8.3% of post-COVID-19 patients with persistent cardiac symptoms. CMR imaging abnormalities (i.e., LGE lesions and active myocarditis) were observed both in previously hospitalized and non-hospitalized post-COVID-19 patients. Interestingly, the extent of lung involvement, as shown on chest CT scans, in the acute phase of COVID-19 pneumonia was unrelated to the prevalence of non-ischemic cardiac injury in our study participants. However, we found that patients with more severe COVID-19 pneumonia assessed according to the clinical criteria were more likely to develop non-ischemic cardiac injury.

Notably, hs-cTnI, NT-pro-BNP, and CRP concentrations were within normal limits in the majority of our study participants. Additionally, NT-pro-BNP and CRP concentrations did not differ between post-COVID-19 patients with and without non-ischemic cardiac injury. Although hs-cTnI concentrations were statistically higher in our post-COVID-19 patients with non-ischemic cardiac injury, only two study participants presented with elevated hs-cTnI concentrations, and in fact, in both of them, we found neither active myocarditis nor LGE lesions when using CMR.

### 4.2. COVID-19 Myocarditis

Several studies have reported delayed myocarditis cases after the initial COVID-19 infection [18,19]. It is well documented that the long-term prognosis following myocarditis varies greatly depending on the patient’s clinical manifestation [20]. Previous data indicated that in the long follow-up of patients recovered from non-COVID-19 myocarditis, the incidence of heart failure hospitalization was 6 to 8% [21]. Thus, it seems crucial to identify patients with persistent cardiac symptoms indicating non-ischemic cardiac injury (such as chest pain, palpitation, dyspnea, fatigue) after recovery from COVID-19 infection.

Importantly, the prevalence of myocarditis in patients with persistent cardiac symptoms after recovery from COVID-19 infection has not been precisely determined so far. Moreover, it is also unclear whether the initial COVID-19 severity, necessity of hospitalization, and different treatment strategies affect the CMR findings after the acute phase of SARS-CoV-2 infection.

Puntmann et al. studied a population of 100 unselected individuals who recently recovered from COVID-19 pneumonia, and demonstrated non-ischemic cardiac injury using CMR in 78% of the population, with ongoing myocardial inflammation diagnosed in 60% of the investigated patients [9]. In this study, most of the analyzed patients recovered at home (67%), and were either asymptomatic or presented with mild to moderate COVID-19 pneumonia (63%). Moreover, CMR findings were irrespective of pre-existing conditions and the overall course of COVID-19 infection. Importantly, this study used different definitions of key endpoints than our trial. Discrepancies between these observations and our cohort may be partly due to differences between the investigated populations. The present study included only patients with persistent cardiac symptoms, while the previous research analyzed unselected individuals. Additionally, a shorter median time interval between the diagnosis of COVID-19 pneumonia and CMR examination in our research than in a study by Puntmann et al. (41 (25; 61) vs. 71 (64; 92) days) might also affect the results.

Tissue characterization techniques showed that the most prevalent imaging abnormality in our study was LGE, with similar frequency in hospitalized and non-hospitalized patients. A previous study by Gulati et al. indicated an association between LGE and mortality in patients with non-ischemic cardiomyopathies [22]. The extent of LGE is a known risk factor for mortality after myocardial infarction [23]. However, LGE detected in the acute phase of myocarditis may not necessarily be sustained over time [24]. It is likely that some LGE areas detected in post-COVID-19 patients may disappear in the follow-up CMR examination. LGE by itself is also insufficient to differentiate fibrosis from persistent inflammation (healed from active myocarditis). The intensity of LGE changes during the acute and healing phase of myocarditis. For optimal visualization, CMR imaging should be performed soon after the acute phase. Some data suggest that in myocarditis, contrast enhancement decreases and might not be visualized after the first 2 weeks from symptom onset [25]. The overall location of LGE lesions in our research was similar to other forms of viral myocarditis, which frequently affects the inferior and inferolateral segments of the left ventricle. However, it should be noted that in a proportion of our patients, these lesions can be unrelated to the recent SARS-CoV-2 infection. Interestingly, unrecognized myocardial scarring was reported by several authors (up to 13 % of patients without history of myocardial infarction in one study [26]). Consequently, it seems likely that unrecognized non-ischemic LGE would be present in a proportion of patients undergoing CMR for other reasons, unrelated to myocarditis. Even though the assessment of unrecognized non-ischemic lesions may be more problematic than the assessment of ischemic scarring, several studies reported specifically on the prevalence of non-ischemic LGE other than right ventricular insertion points in endurance athletes and healthy, physically active adults. Previously unrecognized non-ischemic LGE was detected in 4% to 10% of the healthy persons [27,28]. This needs to be taken into consideration in the interpretation of our results and the results of other cross-sectional studies published to date.

In a recent study by Chen et al. [29], COVID-19 patients were scanned during the acute phase (3–8 days). Native T1, T2, and ECV values were higher than in our study. The obtained results suggest that in COVID-19-related myocarditis acute or diffuse injury, as shown by increased T1, T2, and ECV without LGE, may be more prevalent than localized LGE itself. In light of these findings, and contrary to the well-established significance of LGE in viral myocarditis, in COVID-19-related myocarditis, more subtle myocardial alterations may be detected by newer quantitative techniques such as parametric mapping. In our study, no significant differences in T1, T2, and ECV values were between the compared subgroups (i.e., hospitalized and non-hospitalized post COVID-19 patients). This observation suggests that severity and the overall course of COVID-19 do not necessarily contribute to cardiac injury.

Similar findings were reported by Ng et al. [30] in a case series of 16 patients who underwent CMR at a mean of 56 days post recovery from COVID-19 pneumonia. This study exclusively enrolled patients either with elevated cardiac troponin concentration or with ECG changes noted during the acute phase of the illness. Indeed, non-ischemic cardiac injury consistent with active or healed myocarditis was demonstrated in the majority of study participants, and 19% of them only had elevated T1/T2 without localized LGE. It should be emphasized that the identification of specific tissue characteristics in the early stage of COVID-19 myocarditis would require further investigation.

### 4.3. Left and Right Ventricular Function

In total, 47 of 121 patients (38.8%) in our study had impaired left ventricular systolic function (defined as LVEF < 57%), while non-ischemic cardiac injury was confirmed in more than half of participants. Similar observations had been reported by Huang et al. for a group of patients recently recovered from COVID-19 [10]. The authors suggest it may be a consequence of the relatively early stage of non-ischemic cardiac injury during the CMR procedure, because myocardial tissue remodeling occurs earlier than functional remodeling in the left ventricle.

In our post-COVID-19 cohort, both RVEDV and the prevalence of reduced RVEF was significantly higher in hospitalized vs. non-hospitalized patients. Previous studies have demonstrated acute right ventricle dysfunction due to COVID-19 pneumonia and respiratory failure [31,32]. Moreover, COVID-19-related pulmonary thrombosis involving pulmonary microvessels may lead to an increase in pulmonary vascular resistance and probably an increase in mortality rate [33]. Paternoster et al., in their meta-analysis of 1450 patients (half of them were invasively ventilated), reported high mortality rates among patients with COVID-19 pneumonia requiring respiratory support and with right ventricular dysfunction, dilatation, or pulmonary hypertension [34]. Such abnormalities may be more likely to affect patients requiring hospitalization with more severe respiratory symptoms.

### 4.4. Management of Post-COVID-19 Cardiac Complications

A wide range of post-COVID-19 complications (e.g., myocarditis, myocardial infarction, right ventricular dysfunction/heart failure, and arrhythmias) have been reported so far [35]. Their underlying pathomechanisms are still poorly understood. In general, the management of these complications does not differ from the management of corresponding non-post-COVID-19 cardiac diseases. A recent expert consensus on the management of patients recovered from COVID-19 pneumonia and presenting with cardiac symptoms recommends an initial diagnostic approach comprising of basic laboratory tests, including cardiac troponin, an electrocardiogram (ECG), an echocardiogram, ambulatory ECG Holter monitoring, chest imaging, and/or pulmonary function tests (e.g., spirometry) [36]. Briefly, in patients with elevated cardiac troponin and/or ECG abnormalities indicating myocarditis and/or echocardiographic abnormalities, cardiology consultation is suggested. Furthermore, CMR examination is recommended in hemodynamically stable patients with suspected myocarditis. Importantly, antiviral treatment (e.g., therapy with remdesivir) should be restricted to patients with active SARS-CoV-2 infection.

### 4.5. Study Strengths and Limitations

The present study has several strengths. First of all, we exclusively included patients remaining symptomatic in the early period after the COVID-19 recovery. Secondly, to the authors’ knowledge, it is the first study comparing the prevalence of non-ischemic cardiac injury in hospitalized and non-hospitalized patients with COVID-19 pneumonia. Thirdly, a majority of patients in the study group underwent lung CT in the acute phase of COVID-19 pneumonia. Due to this fact, we were able to investigate a possible relationship between the extent of the lung involvement in the acute phase of COVID-19 pneumonia and subsequent CMR non-ischemic cardiac injury. Fourthly, we applied cardiac parametric mapping for detailed tissue characterization. Finally, all CMR examinations were independently assessed by two experts in the field.

However, some limitations of our study should be acknowledged. We conducted a single-center study with a moderate sample size. Furthermore, our study lacks a control group. Additionally, our study does not provide any insight into the type and extent of myocardial injury during the acute phase of COVID-19 pneumonia. We also did not investigate the SARS-CoV-2 genotype in the study participants. Future, adequately powered studies with follow-up periods and repeated CMR imaging are necessary to more completely evaluate non-ischemic cardiac injury in post-COVID-19 patients.

## 5. Conclusions

In our study, more than half of patients presenting with cardiac symptoms after recent COVID-19 pneumonia recovery had CMR imaging abnormalities indicating non-ischemic cardiac injury. The most common finding was LGE, while active myocarditis was detected in the minority of patients. The majority of LGE lesions were located in the left ventricle at inferior and inferolateral segments at the base. CMR imaging abnormalities (i.e., LGE lesions and active myocarditis) were observed both in previously hospitalized and non-hospitalized post-COVID-19 patients. Importantly, CMR imaging revealed lower RVEF and more frequent presence of reduced RVEF, together with higher RVEDV, in hospitalized vs. non-hospitalized post-COVID-19 patients. Further research is needed to define the long-term cardiovascular consequences of COVID-19 infection and the optimal management of patients with suspected post-COVID-19 non-ischemic cardiac injury.

## Figures and Tables

**Figure 1 biology-11-01848-f001:**
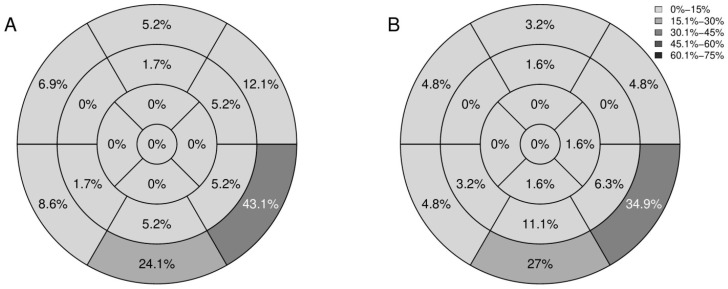
Bull’s eye plots of the distribution of LGE according to the 17-segment AHA classification: rates of segments with LGE in hospitalized (**A**) and non-hospitalized (**B**) post-COVID-19 patients. AHA, American Heart Association; COVID-19, coronavirus disease 2019; LGE, late gadolinium enhancement.

**Figure 2 biology-11-01848-f002:**
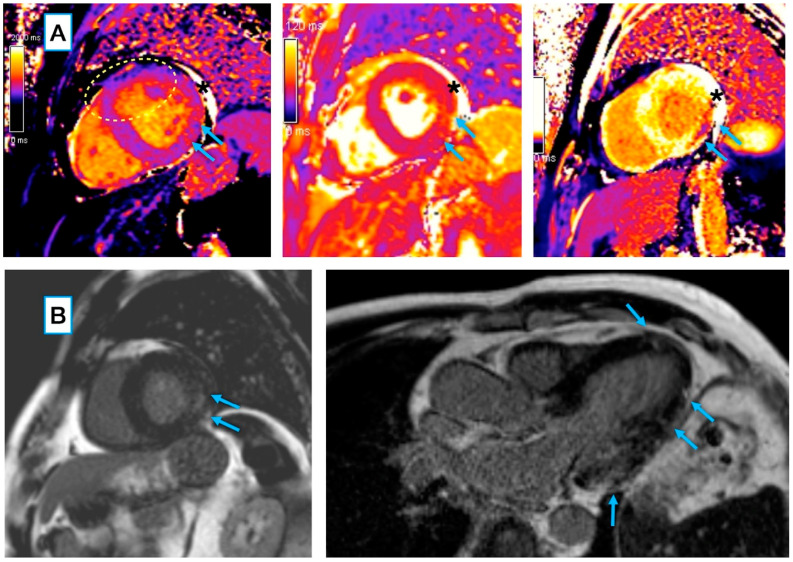
CMR features of myocarditis in a 68-year-old male patient 40 days post COVID-19 pneumonia: (**A**) Locally increased native T1 (**left**) and T2 (**middle**) relaxation times in a mid-ventricular short axis slice, matched by markedly shortened post-contrast T1 (**right**), showing an area of acute injury/ongoing inflammation (arrows). Global T2 relaxation time was normal at 48 ms (the institutional reference range: 39–49 ms), with local elevation in the basal inferolateral segment (arrows; segmental ROI average T2 = 57 ms). Global T1 value was slightly elevated at 1083 ms (the institutional reference range: 951–1035 ms), with a more pronounced increase in the basal inferolateral segment (arrows; segmental ROI average T1 = 1121 ms). The dashed oval marks an area of the artifact related to suboptimal motion correction that was not included in the measurement. * An asterisk marks small pericardial effusion. (**B**, **left**) A corresponding intramyocardial area of LGE, suggestive of at least some extent of inflammatory necrosis/fibrosis (arrows) within the zone of acute injury. (**B**, **right**) In a three-chamber long axis slice, the LGE pattern also includes subtle subepicardial spots along the inferolateral wall, as well as in the apical septal segment (arrows). CMR, cardiac magnetic resonance; LGE, late gadolinium enhancement.

**Figure 3 biology-11-01848-f003:**
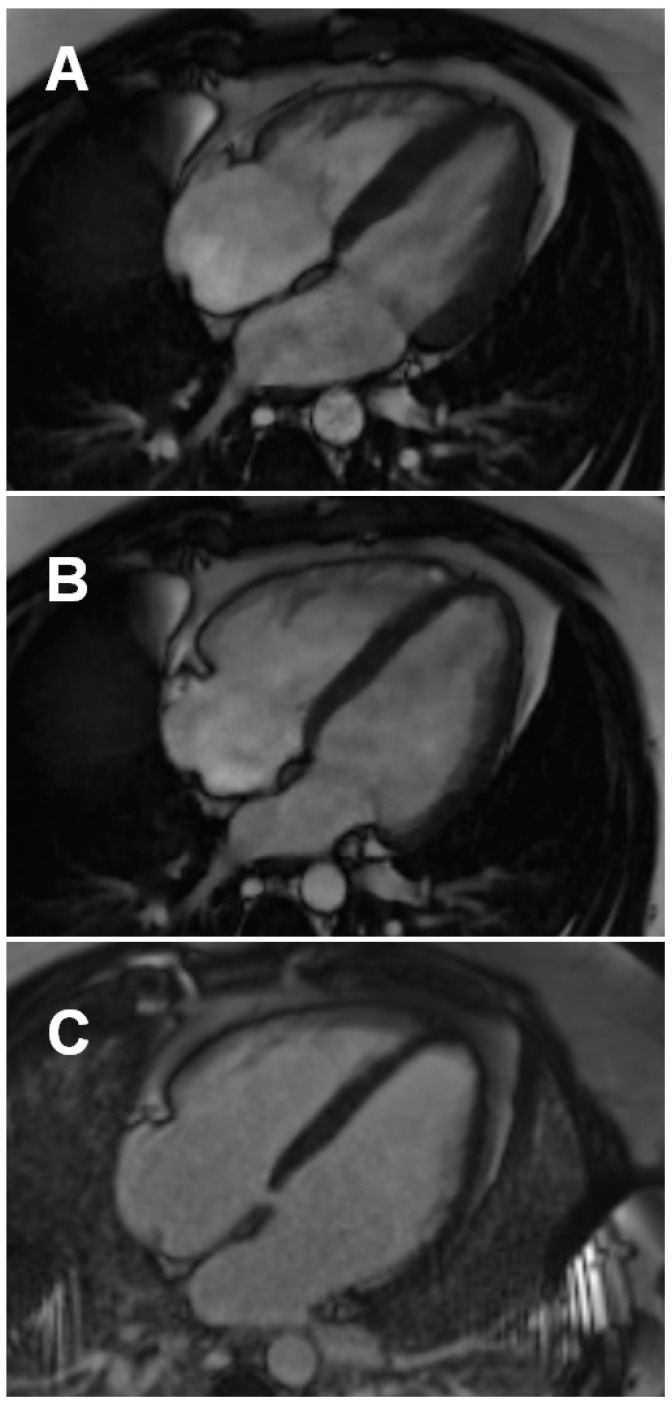
An example of right ventricular enlargement and mild systolic dysfunction in a 26-year-old hospitalized male patient 22 days post COVID-19 pneumonia: Four-chamber systolic (**A**) and four-chamber diastolic (**B**) still frames from routine balanced steady state free precession cine sequence (RVEDV 258 mL; mildly reduced RVEF of 39%). (**C**) No late gadolinium enhancement is visible in the myocardium in a four-chamber PSIR sequence acquired 10 min post contrast injection (Gadovist 0.1 mmol/kg, Bayer, Leverkusen, Germany). Images acquired on Magnetom Sola 1.5T, Siemens, Erlangen, Germany. PSIR, phase-sensitive inversion recovery; RVEDVI, right ventricular end-diastolic volume index; RVEF, right ventricular ejection fraction.

**Table 1 biology-11-01848-t001:** Demographic, clinical, and laboratory characteristics of the study participants.

	All Post-COVID-19 Patients (n = 121)	Hospitalized Post-COVID-19 Patients (n = 58)	Non-Hospitalized Post-COVID-19 Patients (n = 63)	*p*-Value (Hospitalized vs. Non-Hospitalized Post-COVID-19 Patients)
Age, years; Me (IQR)	46 (40; 57)	56.5 (44; 66)	44 (37; 47)	<0.001 ^w^
Female gender; n (%)	55 (45.5)	18 (31.0)	37 (58.7)	0.004 ^c^
Time between COVID-19 diagnosis and CMR examination, days; Me (IQR)	41 (25; 61)	28 (20; 43)	47 (34; 75)	<0.001 ^w^
BMI, kg/m^2^; Me (IQR)	27.2 (25.2; 30.0)	27.9 (26.4; 31.1)	26.3 (23.5; 29.3)	0.008 ^w^
Obesity; n (%)	33 (27.3)	19 (32.8)	14 (22.2)	0.237 ^c^
Hypertension; n (%)	36 (29.8)	24 (41.4)	12 (19.1)	0.013 ^c^
Diabetes; n (%)	15 (12.4)	13 (22.4)	2 (3.2)	0.003 ^c^
Hyperlipidemia; n (%)	32 (26.4)	13 (22.4)	19 (30.2)	0.448 ^c^
Chronic kidney disease; n (%)	2 (1.6)	1 (1.7)	1 (1.6%)	1 ^f^
Chronic respiratory disease *; n (%)	16 (13.2)	11 (19.0)	5 (7.9)	0.128 ^f^
Clinical severity of COVID-19 pneumonia; n (%)				
Mild illness	64 (52.9)	1 (1.7)	63 (100)	<0.001 ^f^
Moderate illness	14 (11.6)	14 (24.1)	0 (0)	
Severe illness	40 (33.0)	40 (69.0)	0 (0)	
Critical illness	3 (2.5)	3 (5.2)	0 (0)	
Extent of the involved lung tissue on computed tomography, %; Me (IQR)	N/A	22.7 (9.8; 35.0)	NR	N/A
Laboratory parameters assessed on the day of CMR examination				
White blood cell count (×10^3^/μL); Me (IQR)	7.0 (5.6; 8.3)	7.0 (5.5; 8.3)	7.0 (6.0; 8.2)	0.829 ^w^
Lymphocyte count (×10^3^/μL); Me (IQR)	2.2 (1.7; 2.5)	2.0 (1.7; 2.5)	2.2 (1.7; 2.6)	0.485 ^w^
Hemoglobin, g/dL; Me (IQR)	14.2 (13.4; 15.1)	14.3 (13.7; 15.3)	14.0 (13.3; 15.0)	0.416 ^t^
CRP, mg/L; Me (IQR)	2.1 (1.0; 4.2)	3.3 (1.7; 6.8)	1.3 (1.0; 2.8)	<0.001 ^w^
Creatinine, mg/dL; Me (IQR)	0.80 (0.74; 0.89)	0.82 (0.75; 0.94	0.78 (0.74; 0.87)	0.058 ^w^
hs-cTnI, ng/mL; Me (IQR)	0.002 (0.002; 0.002)	0.002 (0.002; 0.003)	0.002 (0.002; 0.002)	0.114 ^w^
NT-proBNP, pg/mL; Me (IQR)	56 (36; 100)	69 (41; 122)	49 (34; 80)	0.063 ^w^
Predominant cardiac complaint after recovery from COVID-19 pneumonia; n (%)				
Chest pain	10 (8.3)	1 (1.7)	9 (14.3)	0.011 ^f^
Palpitations	6 (4.9)	1 (1.7)	5 (7.9)	
Dyspnea	53 (43.8)	31 (53.5)	22 (34.9)	
Fatigue	52 (43.0)	25 (43.1)	27 (42.9)	

* Bronchial asthma or obstructive sleep apnea; ^c^ Pearson’s chi-square test with Yates’ continuity correction; ^f^ Fisher’s exact test for count data; ^w^ Wilcoxon rank sum test with continuity correction; ^t^ Student’s *t*-test. BMI, body mass index; COVID-19, coronavirus disease 2019; CMR, cardiac magnetic resonance; CRP, C-reactive protein; hs-cTnI, high-sensitivity cardiac troponin I; IQR, interquartile range; Me, median; N/A, not applicable; NR, not reported; NT-proBNP, N-terminal pro-B-type natriuretic peptide.

**Table 2 biology-11-01848-t002:** CMR characteristics of the study participants.

	All Post-COVID-19 Patients (n = 121)	Hospitalized Post-COVID-19 Patients (n = 58)	Non-Hospitalized Post-COVID-19 Patients (n = 63)	*p*-Value (Hospitalized vs. Non-Hospitalized Post-COVID-19 Patients)
Non-ischemic cardiac injury ^#^; n (%)	64 (54.5)	35 (60.3)	29 (46.0)	0.163 ^c^
LGE lesion; n (%)	63 (52.1)	35 (60.3)	28 (44.4)	0.117 ^c^
Active myocarditis; n (%)	10 (8.3)	6 (10.3)	4 (6.3)	0.517 ^f^
LVEF, %; Me (IQR)	59 (55; 63)	59 (53; 63)	60 (55; 63)	0.198 ^w^
Reduced LVEF *; n (%)	47 (38.8)	24 (41.4)	23 (36.5)	0.717 ^c^
LVESV, mL; Me (IQR)	62 (51; 77)	63 (51; 82)	62 (51; 70)	0.297 ^w^
LVEDV, mL; Me (IQR)	156 (135; 186)	148.5 (133; 180)	145 (132; 170)	0.301 ^w^
LVSV, mL; Me (IQR)	86 (76; 101)	87 (79; 100)	86 (74; 104)	0.775 ^w^
Myocardial mass, g; Me (IQR)	113 (93; 141)	122 (103; 147)	103 (87; 120)	<0.001 ^w^
Global T1, ms; Me (IQR)	1012 (994; 1031)	1017 (1001; 1033)	1009.5 (991; 1024)	0.133 ^t^
Global T1 > 1035 ms; n (%)	22 (18.8)	13 (23.6)	9 (14.5)	0.306 ^c^
Global T2, ms; Me (IQR)	46 (44; 48)	47 (45; 48)	46 (44; 48)	0.411 ^w^
Global T2 > 49 ms; n (%)	12 (10.1)	9 (15.8)	3 (4.8)	0.093 ^c^
Global ECV, %; Me (IQR)	25 (24; 28)	25 (24; 28)	25 (24; 27)	0.687 ^w^
RVEDV, mL; Me (IQR)	137 (111; 164)	141 (123; 166)	130 (107; 154)	0.029 ^w^
RVEF, %; Me (IQR)	52 (47; 56)	49.5 (44; 54)	53 (50; 58)	0.001 ^w^
Reduced RVEF **; n (%)	56 (46.3)	35 (60.3)	21 (33.3)	0.005 ^c^
Pericardial effusion; n (%)	2 (1.7%)	1 (1.7%)	1 (1.6%)	1 ^f^

^#^ Non-ischemic cardiac injury indicates the primary study endpoint (i.e., LGE lesion and/or CMR features of active myocarditis); * <57%; ** <52% in men and <51% in women; ^c^ Pearson’s chi-square test with Yates’ continuity correction; ^f^ Fisher’s exact test for count data; ^w^ Wilcoxon rank sum test with continuity correction; ^t^ Student’s *t*-test. CMR, cardiac magnetic resonance; COVID-19, coronavirus disease 2019; ECV, extracellular volume; IQR, interquartile range; LGE, late gadolinium enhancement; LVEF, left ventricular ejection fraction; LVEDV, left ventricular end-systolic volume; LVESV, left ventricular end-systolic volume; LVSV, left ventricular stroke volume; Me, median; N/A, not applicable; NR, not reported; RVEDV, right ventricular end-diastolic volume; RVEF, right ventricular ejection fraction.

**Table 3 biology-11-01848-t003:** Comparison of demographic, clinical, and laboratory characteristics between post-COVID-19 patients with and without non-ischemic cardiac injury. Non-ischemic cardiac injury indicates the presence of LGE and/or CMR features of active myocarditis.

	Post-COVID-19 Patients with Non-Ischemic Cardiac Injury Using CMR (n = 64)	Post-COVID-19 Patients without Non-Ischemic Cardiac Injury Using CMR (n = 57)	*p*-Value for the Comparison between the Groups
Age, years; Me (IQR)	48 (44; 62)	44 (37; 50)	0.008 ^w^
Female gender; n (%)	26 (40.6)	29 (50.9)	0.343 ^c^
Time between COVID-19 diagnosis and CMR examination, days; Me (IQR)	41 (26; 48)	42 (25; 67	0.459 ^w^
BMI, kg/m^2^; Me (IQR)	28.1 (25.4; 30.8)	27 (24.3; 30.0)	0.181 ^w^
Obesity; n (%)	19 (29.7)	14 (24.6)	0.669 ^c^
Hypertension; n (%)	24 (37.5)	12 (21.1)	0.076 ^c^
Diabetes; n (%)	11 (17.2)	4 (7)	0.156 ^c^
Hyperlipidemia; n (%)	24 (37.5)	8 (14.0)	0.007 ^c^
Chronic kidney disease; n (%)	1 (1.6)	1 (1.8)	1 ^f^
Chronic respiratory disease *; n (%)	7 (10.9)	9 (15.8)	0.605 ^c^
Clinical severity of COVID-19 pneumonia; n (%)			
Mild illness	30 (46.9%)	34 (59.6%)	
Moderate illness	5 (7.8%)	9 (15.8%)	0.035 ^f^
Severe illness	28 (43.8%)	12 (21.1%)	
Critical illness	1 (1.6%)	2 (3.5%)	
Extent of the involved lung tissue on computed tomography (%) ^; Me (IQR)	24 (11; 49)	22 (10; 30)	0.309 ^w^
White blood cell count (×10^3^/μL); Me (IQR)	7.0 (5.5; 8.3)	7.0 (5.8; 8.2)	0.939 ^t^
Lymphocyte count (×10^3^/μL); Me (IQR)	2.0 (1.7; 2.5)	2.3 (1.6; 2.5)	0.869 ^t^
Hemoglobin, g/dL; Me (IQR)	14.4 (13.3; 15.4)	13.9 (13.4; 14.8)	0.161 ^t^
CRP, mg/L; Me (IQR)	2.4 (1.0; 4.2)	1.9 (1.0; 4.1)	0.989 ^w^
Creatinine, mg/dL; Me (IQR)	0.82 (0.76; 0.92)	0.79 (0.73; 0.86)	0.023 ^w^
hs-cTnI, ng/mL; Me (IQR)	0.002 (0.002; 0.003)	0.002 (0.002; 0.002)	0.009 ^w^
NT-proBNP, pg/mL; Me (IQR)	58 (40; 106)	52 (34; 96)	0.549 ^w^
Predominant cardiac complaint after recovery from COVID-19 pneumonia; n (%)			
Chest pain	3 (4.7)	7 (12.3)	
Palpitations	2 (3.1)	4 (7)	0.237 ^f^
Dyspnea	32 (50)	21 (36.8)	
Fatigue	27 (42.2)	25 (43.9)	

* Bronchial asthma or obstructive sleep apnea; ^ data were obtained exclusively for hospitalized patients; ^c^ Pearson’s chi-square test with Yates’ continuity correction; ^f^ Fisher’s exact test for count data; ^w^ Wilcoxon rank sum test with continuity correction; ^t^ Student’s *t*-test. COVID-19, coronavirus disease 2019; BMI, body mass index; CMR, cardiac magnetic resonance; CRP, C-reactive protein; hs-cTnI, high-sensitivity cardiac troponin I; IQR, interquartile range; LGE, late gadolinium enhancement; Me, median; NT-proBNP, N-terminal pro-B-type natriuretic peptide.

**Table 4 biology-11-01848-t004:** Comparison of CMR characteristics between post-COVID-19 patients with and without non-ischemic cardiac injury. Non-ischemic cardiac injury indicates the presence of LGE and/or CMR features of active myocarditis.

	Post-COVID-19 Patients with Non-Ischemic Cardiac Injury Using CMR (n = 64)	Post-COVID-19 Patients without Non-Ischemic Cardiac Injury Using CMR (n = 57)	*p*-Value for the Comparison between the Groups
LGE lesion; n (%)	63 (98.4)	0 (0)	<0.001 ^c^
Active myocarditis; n (%)	10 (15.6)	0 (0)	0.001 ^f^
LVEF, %; Me (IQR)	57 (52; 62)	61 (56; 64)	<0.001 ^t^
Reduced LVEF *; n (%)	32 (50%)	15 (26.3%)	0.013 ^c^
LVESV, mL; Me (IQR)	64 (53; 83)	59 (49; 68)	0.046 ^w^
LVEDV, mL; Me (IQR)	147 (134; 177)	148 (128; 170)	0.289 ^w^
LVSV, mL; Me (IQR)	85 (76; 101)	89 (76; 102)	0.605 ^w^
Myocardial mass, g; Me (IQR)	117 (99; 143)	103 (87; 132)	0.033 ^w^
Global T1, ms; Me (IQR)	1019 (996; 1037)	1008 (994; 1024)	0.097 ^w^
Global T1 > 1035 ms; n (%)	16 (25.8)	6 (10.9)	0.068 ^c^
Global T2, ms; Me (IQR)	47 (44; 49)	46 (45; 48)	0.336 ^w^
Global T2 > 49 ms; n (%)	9 (14.1)	3 (5.5)	0.211 ^c^
Global ECV, %; Me (IQR)	26 (24; 28)	25 (24; 27)	0.326 ^w^
RVEDV, mL; Me (IQR)	143 (117; 165)	127 (107; 158)	0.098 ^t^
RVEF, %; Me (IQR)	52 (46; 55)	54 (49; 58)	0.236 ^w^
Reduced RVEF **; n (%)	30 (46.9)	26 (45.6)	1 ^c^
Pericardial effusion; n (%)	0 (0%)	2 (3.5%)	0.220 ^f^

* Bronchial asthma or obstructive sleep apnea; ^c^ Pearson’s chi-square test * < 57%, ** < 52% in men and <51% in women; ^c^ Pearson’s chi-square test with Yates’ continuity correction; ^f^ Fisher’s exact test for count data; ^w^ Wilcoxon rank sum test with continuity correction; ^t^ Student’s *t*-test. CMR, cardiac magnetic resonance; COVID-19, coronavirus disease 2019; ECV, extracellular volume; IQR, interquartile range; LGE, late gadolinium enhancement; LVEF, left ventricular ejection fraction; LVEDV, left ventricular end-systolic volume; LVESV, left ventricular end-systolic volume; LVSV, left ventricular stroke volume; LGE, late gadolinium enhancement; Me, median; N/A, not applicable; NR, not reported; RVEDV, right ventricular end-diastolic volume; RVEF, right ventricular ejection fraction.

## Data Availability

The data presented in this study are available on request from the corresponding author.

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
