# Peer review of "Cardiac Magnetic Resonance Findings in Patients Recovered from COVID-19 Pneumonia and Presenting with Persistent Cardiac Symptoms: The TRICITY-CMR Trial"

_biology, 2022, doi:10.3390/biology11121848_

Round 1
Reviewer 1 Report
(A) GENERAL COMMENTS:
Authors present a single-center, cross-sectional study on patients recovered from COVID-19 infection with persistent cardiac symptoms to assess the prevalence of non-ischemic cardiac injury assessed in cardiac magnetic resonance with contrast (CMR). Studied group was compared with patients referred in the past to CMR with suspicion of non-COVID-19 myocarditis. Authors interpreted presence of LGE or signs of acute myocarditis as cardiac involvement of COVID-19 infection in studied group. For more detailed comparisons, convalescents from COVID-19 infection were divided into subgroups of patients that required or not required hospitalization during acute phase of COVID-19 pneumonia. Presenting study shows that 53% of patients recovered from COVID-19 infection had non-ischemic cardiac injury in CMR (mainly manifested as the presence of LGE without edema- 54/64 patients, 84%), whereas in the control group the ischemic cardiac injury was presented in 85% of patients (also mainly presented as the presence of LGE without edema- 14/17 patients, 82%). No significant difference in LGE, T1-, T2- mappings and ECV was observed between hospitalized and non-hospitalized group post-COVID-19 patients, however hospitalized subgroup had higher LV myocardial mass and RVEDV, lower right ventricular ejection fraction and more frequent had reduced RFEF.
The manuscript is generally well written, presents unique and very interesting CMR, laboratory and clinical data, good figures and tables, and topic is of high importance and may be of readers’ interest, however the main text of manuscript needs some improvement according to the detailed comments below.
(B) SPECIFIC RECOMMENDATIONS FOR REVISION:
* MAJOR *:
1. Materials and Methods:
a) Study Design and Population, line 108: Patients were recruited at the outpatient post-COVID-19 clinic. Please precise who/what profile of patients were referred to have a visit in this clinic? For example: all patients after COVID-19 infection referred by a family doctor or only those who had persistent symptoms/ problems after COVID-19 infection?
b) Study Design and Population, lines 110-112: Please precise the exact time frame for "recently" (from RT-PCR positive swab test) in inclusion criteria (i).
c) Study Design and Population, lines 137-142: Control group is one of the main concern in the presenting study.
- How Authors justify their choice of control group consisted with patients with suspect myocarditis? It seems that the control group were sicker than the studied group. It would more valuable to have sex- and age matched control group of patients recovered from COVID-19 infection without any cardiac symptoms or even a healthy, local population without any cardiac problems (healthy control group) to evaluate a prevalence of non-ischemic pattern of LGE in such defined control group/ population. In that case we would be aware what is the frequency of LGE as incidental findings. Without such defined control group it is impossible to discuss if LGE that were found in post- COVID-19 patients were somehow related to COVID-19 infection or were just incidental findings.
- Authors did not explain if in control group were patients with suspected acute myocarditis and positive troponin level/MINOCA or patients with some cardiac symptoms and referred for regular check-up with MR to confirm/exclude (subacute/chronic) myocarditis in the ambulatory settings. Also it is important to mention what was the timeframe from the beginning of symptoms to CMR (2 weeks or more?).
- The control group was only sex-matched. Was it also age-matched, and if not could Authors explain the reason of only sex-matched control group?
- Importantly, Authors did not mention any results and conclusions regarding comparison with control group in abstract. There is also no final conclusion in the main text of manuscript connected with control group, therefore there is a doubt if such defined control group is really needed in the presenting study. Please re-think the importance of the comparison with control group, add additional conclusion or change the control group. (Paragraph: Abstract and 5. Conclusions).
d) Study endpoints, lines 158-160. Please explain more precisely "T1" and "T2" in method section as the relaxation time of myocardium in T1-/ T2- mapping sequence. Please add information that ECV was measured only for studied group based on the calculation using hematocrit from the day of the CMR study.
e) CMR Scanning Protocol and Image Analysis, line 166: Technical information about a coil that was used in this study is missing (e.g. how many channels had the coil?).
f) CMR Scanning Protocol and Image Analysis, line 170: Information about what views/slices of left ventricle were used for LGE and mappings sequence is missing, as well as technical details of what sequence for mappings and LGE were used.
2. Figure 1.
This figure probably needs correction, as in the main text of manuscript is written that the most common involved segments were inferior and inferolateral, whereas at this figure are inferoseptal and inferior, so somewhere (in this figure or in the text) must be a mistake.
3. Figure 2.
Lines 349-350: The authors interpreted area of LGE in inferolateral segment in patient with acute myocarditis (with elevated T2 mapping in this zone) as irreversible damage of the myocardium. Authors should delete this statement ("irreversible") or justify their opinion. As authors self-admit later in the discussion section, lines 463-465: "(...) LGE detected in the acute phase of myocarditis does not necessarily result in myocardial fibrosis. (...)". In acute settings, LGE may represent also reactive fibrosis and not necessarily necrosis.
4. Discussion:
a) Major Study Findings, line 409-410 (and general in the whole manuscript): Authors intrerpreted presence of LGE or signs of acute myocarditis as cardiac involvement of COVID-19 infection in studied group. Direct connection between observed CMR changes and COVID-19 infection was not examined in the presenting study and cannot be justify. It cannot be proved that observed cardiac involvement in CMR is related to COVID-19 infection when there was no CMR study before infection or the follow up study and there was no comparison with healthy control group? LGE might be an incidental finding and might be present also after myocarditis in the past/ childhood etc. (as Authors self-admit in the study limitation section: line 530-532). When Authors cannot proved that this findings were related to COVID-19 infection it is better to use a statement "non-ischemic cardiac injury" instead of "post-COVID-19 cardiac involvement".
b) COVID-19 Myocarditis, line 471-472: On the presenting bull-eye model (Figure 1) the most common in the study was inferoseptal and inferior, but not inferolateral. Please explain that difference or correct the figure/ text.
c) Study Strengths and Limitations, lines 530-532: One of the main limitations of the study is that there is no comparison to/reference to the prevalence of "incidental LGE" in similar healthy population/ patients after COVID-19 infection but without cardiac symptoms. This limitation should be emphasized, if Authors cannot add such data. Therefore I would recommend to not use the phrase "cardiac involvement of COVID-19 infection" or similar regarding the study population in the whole manuscript. More correct is an observation that "non-ischemic cardiac injury" was present in studied population. In fact, the relation with COVID-19 were not checked in the presenting study.
** MINOR**:
5. Some spelling mistakes need corrections:
- Line 156: Please remove a full stop after “(<57%).”
- Line 157: Please write “diastolic” instead of systolic in the sentence: “left ventricular end-systolic volume (LVEDV)”.
- Table 1, first row (age): the closing parenthesis is not visible in the table.
Author Response
Dear Editor Ms. Lillie Zhang,
Dear Reviewers,
We appreciate very much your careful review of our manuscript.
We have been able to resolve the issues raised by the reviewers, and have made appropriate changes accordingly.
Please find attached the detailed reply to all comments/suggestions.
Changes made in the manuscript are in red.
We trust that you will find these changes satisfactory and we look forward to hearing from you at your earliest convenience.
Yours sincerely,
on behalf of all authors
Marek Kozinski, MD, PhD
Reviewer No 1
Comment No 1:
(A) GENERAL COMMENTS:
Authors present a single-center, cross-sectional study on patients recovered from COVID-19 infection with persistent cardiac symptoms to assess the prevalence of non-ischemic cardiac injury assessed in cardiac magnetic resonance with contrast (CMR). Studied group was compared with patients referred in the past to CMR with suspicion of non-COVID-19 myocarditis. Authors interpreted presence of LGE or signs of acute myocarditis as cardiac involvement of COVID-19 infection in studied group. For more detailed comparisons, convalescents from COVID-19 infection were divided into subgroups of patients that required or not required hospitalization during acute phase of COVID-19 pneumonia. Presenting study shows that 53% of patients recovered from COVID-19 infection had non-ischemic cardiac injury in CMR (mainly manifested as the presence of LGE without edema- 54/64 patients, 84%), whereas in the control group the ischemic cardiac injury was presented in 85% of patients (also mainly presented as the presence of LGE without edema- 14/17 patients, 82%). No significant difference in LGE, T1-, T2- mappings and ECV was observed between hospitalized and non-hospitalized group post-COVID-19 patients, however hospitalized subgroup had higher LV myocardial mass and RVEDV, lower right ventricular ejection fraction and more frequent had reduced RFEF.
The manuscript is generally well written, presents unique and very interesting CMR, laboratory and clinical data, good figures and tables, and topic is of high importance and may be of readers’ interest, however the main text of manuscript needs some improvement according to the detailed comments below.
Reply: We are very grateful for your in-depth review of our manuscript and excellent comments which helped us to improve the text.
Comment No 2:
- a) Study Design and Population, line 108: Patients were recruited at the outpatient post-COVID-19 clinic. Please precise who/what profile of patients were referred to have a visit in this clinic? For example: all patients after COVID-19 infection referred by a family doctor or only those who had persistent symptoms/ problems after COVID-19 infection?
Reply: Thank you for this comment. In the revised manuscript, we specified that “Study candidates with persistent cardiac symptoms were referred by family doctors, cardiologists, pulmonologists and specialists in infectious diseases from outpatient clinics as well as physicians from dedicated COVID-19 hospital wards from the Pomerania region. Information on the activity of our outpatient post-COVID-19 cardiology clinic was widely disseminated by local print, broadcasting and internet media.”
Comment No 3:
- b) Study Design and Population, lines 110-112: Please precise the exact time frame for "recently" (from RT-PCR positive swab test) in inclusion criteria (i).
Reply: We agree that this information should be added to the text. Therefore, when listing study inclusion criteria, we wrote “Study inclusion criteria comprised: i) SARS-CoV-2 infection confirmed using reverse transcription-polymerase chain reaction (RT-PCR) swab test within one month before the occurrence of new cardiac symptoms, (…)”
Comment No 4:
- c) Study Design and Population, lines 137-142: Control group is one of the main concern in the presenting study.
- How Authors justify their choice of control group consisted with patients with suspect myocarditis? It seems that the control group were sicker than the studied group. It would more valuable to have sex- and age matched control group of patients recovered from COVID-19 infection without any cardiac symptoms or even a healthy, local population without any cardiac problems (healthy control group) to evaluate a prevalence of non-ischemic pattern of LGE in such defined control group/ population. In that case we would be aware what is the frequency of LGE as incidental findings. Without such defined control group, it is impossible to discuss if LGE that were found in post- COVID-19 patients were somehow related to COVID-19 infection or were just incidental findings.
Reply: We would like to thank the Reviewer for their insightful comment. After careful reconsideration, we decided to withhold the non-COVID control group, focus on the findings derived from the post-COVID subgroups, and to refer our findings to the available data from literature regarding incidental non-ischemic LGE findings. Appropriate passage was now added to the Discussion (marked in red). We wrote in the revised discussion section:
“The overall location of LGE lesions in our research was similar to other viral myocarditis, which frequently affects the inferior and infero-lateral segments of the left ventricle. However, it should be noted that in a proportion of our patients, these lesions can be unrelated to the recent SARS-Cov2 infection. Interestingly, unrecognized myocardial scar was reported by several authors (up to 13 % of patients without history of MI in one study [26]). Consequently, it seems likely that unrecognized non-ischemic LGE would be present in a proportion of patients undergoing CMR for other reasons, unrelated to myocarditis. Even though the assessment of unrecognized non-ischemic lesions may be more problematic than the assessment of ischemic scar, several studies reported specifically on the prevalence of non-ischemic LGE other than right ventricular insertion points in endurance athletes and healthy physically active adults. Previously unrecognized non-ischemic LGE was detected in 4% to 10% of the healthy persons [27,28]. This needs to be taken into consideration in the interpretation of our results and the results of other cross-sectional studies published to date.”
- Authors did not explain if in control group were patients with suspected acute myocarditis and positive troponin level/MINOCA or patients with some cardiac symptoms and referred for regular check-up with MR to confirm/exclude (subacute/chronic) myocarditis in the ambulatory settings. Also, it is important to mention what was the timeframe from the beginning of symptoms to CMR (2 weeks or more?).
Reply: In parallel with the post-COVID group, these were outpatients with protracted symptoms that might have been related to subacute/chronic myocarditis and the timeframe from the beginning of symptoms to CMR was several weeks (i.e. >2 weeks). However, we agree that COVID-19 pneumonia that our patients had sustained prior to inclusion, may introduce some bias regarding the symptoms in this setting. Hence, we decided to withhold, as stated above. Once again, thank you for this in-depth remark.
- The control group was only sex-matched. Was it also age-matched, and if not could Authors explain the reason of only sex-matched control group?
Reply: Please see our response above.
- Importantly, Authors did not mention any results and conclusions regarding comparison with control group in abstract. There is also no final conclusion in the main text of manuscript connected with control group, therefore there is a doubt if such defined control group is really needed in the presenting study. Please re-think the importance of the comparison with control group, add additional conclusion or change the control group. (Paragraph: Abstract and 5. Conclusions).
Reply: Please see our response above.
Comment No 5:
- d) Study endpoints, lines 158-160. Please explain more precisely "T1" and "T2" in method section as the relaxation time of myocardium in T1-/ T2- mapping sequence. Please add information that ECV was measured only for studied group based on the calculation using hematocrit from the day of the CMR study.
Reply: Thank you. The information has now been added to the respective paragraph. In the revised text, we wrote:
“Additional endpoints included CMR-assessed left ventricular ejection fraction (LVEF), the presence of reduced LVEF (<57%). [15], left ventricular end-systolic volume (LVESV), left ventricular end-systolic volume (LVEDV), left ventricular stroke volume (LVSV), myocardial mass, global longitudinal relaxation time (T1), prolonged global T1 (>1035 ms; the institutional reference range: 951–1035 ms), global transverse relaxation time (T2), prolonged global T2 (>49 ms; the institutional reference range: 39–49 ms), global extracellular volume (ECV) fraction based on the commonly adopted formula incorporating blood hematocrit, right ventricular end-diastolic volume (RVEDV), right ventricular ejection fraction (RVEF), reduced RVEF (<52% and <51% in men and in women, respectively) [15], the presence of pericardial effusion, and computed tomography (CT)-assessed extent of the COVID-19 pneumonia.”
Comment No 6:
- e) CMR Scanning Protocol and Image Analysis, line 166: Technical information about a coil that was used in this study is missing (e.g. how many channels had the coil?).
Reply: Thank you. The information has now been added to the respective paragraph. In the revised text, we wrote:
“All participants underwent CMR examination on 1.5 Tesla MR scanners (Magnetom Area or Magnetom Sola, Siemens AG, Erlangen, Germany, with an 18-element phased array cardiac coil) using standardized imaging protocols.”
Comment No 7:
- f) CMR Scanning Protocol and Image Analysis, line 170: Information about what views/slices of left ventricle were used for LGE and mappings sequence is missing, as well as technical details of what sequence for mappings and LGE were used.
Reply: Thank you. The information has now been added to the respective paragraph. In the revised text, we wrote:
“All participants underwent CMR examination on 1.5 Tesla MR scanners (Magnetom Area or Magnetom Sola, Siemens AG, Erlangen, Germany, with an 18-element phased array cardiac coil) using standardized imaging protocols. Theseincluded long axis and short axis cine series for anatomy and functional assessment, followed by cardiac parametric mapping sequences for longitudinal (T1) and transverse (T2) relaxation time measurement (MOLLI [Modified Look-Locker] sequence] for T1 and a T2-prepared bSSFP sequence for T2, respectively (MyoMaps, Siemens Healthineers, Erlangen, Germany) as well as routine LGE assessment in the long axes and a short axis stack using both a fast single shot bSSFP inversion recovery and a segmented phase-sensitive inversion recovery sequences, performed within 7–15 min after injection of 0.1 mmol/kg of gadobutrol (Gadovist, Bayer AG, Leverkusen, Germany) [16]. The left ventricle was divided into 17 segments according to the American Heart Association (AHA) [17]. All CMR images were evaluated by the same two physicians (a cardiologist and a radiologist, both with long-standing experience in CMR) using commercial software (SyngoVia VB40, Siemens Healthineers, Erlangen, Germany). Any doubts were resolved by a consensus decision after discussion with a third experienced CMR reader. Persons analyzing CMR scans were unaware of the results of laboratory biomarkers.”
Comment No 8:
Figure 1.
This figure probably needs correction, as in the main text of manuscript is written that the most common involved segments were inferior and inferolateral, whereas at this figure are inferoseptal and inferior, so somewhere (in this figure or in the text) must be a mistake.
Reply: We are grateful for finding this mistake. The figure was corrected.
Comment No 9:
Figure 2.
Lines 349-350: The authors interpreted area of LGE in inferolateral segment in patient with acute myocarditis (with elevated T2 mapping in this zone) as irreversible damage of the myocardium. Authors should delete this statement ("irreversible") or justify their opinion. As authors self-admit later in the discussion section, lines 463-465: "(...) LGE detected in the acute phase of myocarditis does not necessarily result in myocardial fibrosis. (...)". In acute settings, LGE may represent also reactive fibrosis and not necessarily necrosis.
Reply: We would like to thank the Reviewer for this insightful comment. Indeed, the issue of LGE in the acute setting is more complex. Therefore, we have modified the text both in the Figure 2 legend and in the discussion, to reflect the fact that even though the LGE areas most commonly contain at least some irreversible inflammatory damage, it does not necessarily show in follow-up CMR when edema completely resolves (depending both on the amount of damaged cardiomyocytes and their distribution in the ventricular wall. Consequently, sparsely distributed tiny foci of inflammatory necrosis are more likely to disappear on the follow-up CMR as they result in spots of fibrosis that may be below the spatial resolution of the standard LGE technique. We sincerely hope this explanation along with the adjustments made to the text will be satisfactory for the Reviewer.
In the revised Figure 2 legend, we wrote:
“CMR features of COVID-19-related myocarditis in a 68-year-old male patient 40 days post pneumonia. A. Locally increased native T1 (left) and T2 (middle) relaxation times in a mid-ventricular short axis slice, matched by markedly shortened post-contrast T1 (right), showing an area of acute injury/ongoing inflammation (arrows). Global T2 relaxation time was normal at 48 ms (the institutional reference range: 39-49 ms), with local elevation in the basal inferolateral segment (arrows; segmental ROI average T2=57 ms). Global T1 value was slightly elevated at 1083 ms (the institutional reference range: 951-1035 ms) with a more pronounced increase in the basal inferolateral segment (arrows; segmental ROI average T1=1121ms). The dashed oval marks an area of the artifact related to suboptimal motion correction that was not included in the measurement. An asterisk marks small pericardial effusion. B, left. A corresponding intramyocardial area of LGE, suggestive of at least some extent of inflammatory necrosis/fibrosis (arrows) within the zone of acute injury. B, right. In a 3-chamber long axis slice, the LGE pattern also includes subtle subepicardial spots along the inferolateral wall, as well as in the apical septal segment (arrows). CMR – cardiac magnetic resonance, LGE – late gadolinium enhancement.”
In the revised discussion, we wrote:
“Tissue characterization techniques showed that the most prevalent imaging abnormality in our study was LGE, with similar frequency in hospitalized and non-hospitalized patients. A previous research by Gulati et al. indicated an association between LGE and mortality in patients with non-ischemic cardiomyopathies [22]. The extent of LGE is a known risk factor for mortality after myocardial infarction [23]. However, LGE detected in the acute phase of myocarditis may not necessarily be sustained over time [24]. It is likely that some LGE areas detected in post-COVID-19 patients may disappear in the follow-up CMR examination. LGE by itself is also insufficient to differentiate fibrosis from persistent inflammation (healed from active myocarditis). The intensity of LGE changes during the acute and healing phase of myocarditis. For optimal visualization CMR imaging should be performed soon after the acute illness. Some data suggest that in myocarditis contrast enhancement decreases and might not be visualized after the first two weeks from symptoms onset [25]. The overall location of LGE lesions in our research was similar to other viral myocarditis, which frequently affects the inferior and infero-lateral segments of the left ventricle. However, it should be noted that in a proportion of our patients, these lesions can be unrelated to the recent SARS-Cov2 infection. Interestingly, unrecognized myocardial scar was reported by several authors (up to 13 % of patients without history of MI in one study [26]). Consequently, it seems likely that unrecognized non-ischemic LGE would be present in a proportion of patients undergoing CMR for other reasons, unrelated to myocarditis. Even though the assessment of unrecognized non-ischemic lesions may be more problematic than the assessment of ischemic scar, several studies reported specifically on the prevalence of non-ischemic LGE other than right ventricular insertion points in endurance athletes and healthy physically active adults. Previously unrecognized non-ischemic LGE was detected in 4% to 10% of the healthy persons [27,28]. This needs to be taken into consideration in the interpretation of our results and the results of other cross-sectional studies published to date.”
Comment No 10:
- Discussion:
- a) Major Study Findings, line 409-410 (and general in the whole manuscript): Authors interpreted presence of LGE or signs of acute myocarditis as cardiac involvement of COVID-19 infection in studied group. Direct connection between observed CMR changes and COVID-19 infection was not examined in the presenting study and cannot be justify. It cannot be proved that observed cardiac involvement in CMR is related to COVID-19 infection when there was no CMR study before infection or the follow up study and there was no comparison with healthy control group? LGE might be an incidental finding and might be present also after myocarditis in the past/ childhood etc. (as Authors self-admit in the study limitation section: line 530-532). When Authors cannot prove that these findings were related to COVID-19 infection it is better to use a statement "non-ischemic cardiac injury" instead of "post-COVID-19 cardiac involvement".
Reply: We are grateful for this comment and we agree with the reviewer. Therefore, "post-COVID-19 cardiac involvement" was changed for "non-ischemic cardiac injury" throughout the manuscript. Additionally, we addressed this comment in the revised text. In the revised discussion, we wrote:
“The overall location of LGE lesions in our research was similar to other viral myocarditis, which frequently affects the inferior and infero-lateral segments of the left ventricle. However, it should be noted that in a proportion of our patients, these lesions can be unrelated to the recent SARS-CoV-2 infection. Interestingly, unrecognized myocardial scar was reported by several authors (up to 13 % of patients without history of myocardial infarction in one study [26]). Consequently, it seems likely that unrecognized non-ischemic LGE would be present in a proportion of patients undergoing CMR for other reasons, unrelated to myocarditis. Even though the assessment of unrecognized non-ischemic lesions may be more problematic than the assessment of ischemic scar, several studies reported specifically on the prevalence of non-ischemic LGE other than right ventricular insertion points in endurance athletes and healthy physically active adults. Previously unrecognized non-ischemic LGE was detected in 4% to 10% of the healthy persons [27,28]. This needs to be taken into consideration in the interpretation of our results and the results of other cross-sectional studies published to date.”
- b) COVID-19 Myocarditis, line 471-472: On the presenting bull-eye model (Figure 1) the most common in the study was inferoseptal and inferior, but not inferolateral. Please explain that difference or correct the figure/ text.
Reply: We are grateful for finding this mistake in Fugure 1. We confirm that LGE lesions were most frequent in the left ventricle at inferior and infero-lateral segments at base. The figure was corrected.
- c) Study Strengths and Limitations, lines 530-532: One of the main limitations of the study is that there is no comparison to/reference to the prevalence of "incidental LGE" in similar healthy population/ patients after COVID-19 infection but without cardiac symptoms. This limitation should be emphasized, if Authors cannot add such data. Therefore, I would recommend to not use the phrase "cardiac involvement of COVID-19 infection" or similar regarding the study population in the whole manuscript. More correct is an observation that "non-ischemic cardiac injury" was present in studied population. In fact, the relation with COVID-19 were not checked in the presenting study.
Reply: We are grateful for this comment and we agree with the reviewer. Therefore, "post-COVID-19 cardiac involvement" was changed for "non-ischemic cardiac injury" throughout the manuscript.
** MINOR**:
- Some spelling mistakes need corrections:
- Line 156: Please remove a full stop after “(<57%).”
Reply: Thank you. It was corrected accordingly.
- Line 157: Please write “diastolic” instead of systolic in the sentence: “left ventricular end-systolic volume (LVEDV)”.
Reply: Thank you. It was corrected accordingly.
- Table 1, first row (age): the closing parenthesis is not visible in the table.
Reply: Thank you. It was corrected accordingly.

Reviewer 2 Report
The manuscript entitled “ Cardiac Magnetic Resonance Findings in Patients Recovered 2 from COVID-19 Pneumonia and Presenting with Persistent 3 Cardiac Symptoms: the TRICITY-CMR Trial” sought to assess the prevalence of cardiac involvement in patients after COVID-19 recovery. The finding of RV involvement is exciting and novel. Below are some minor comments that can further strengthen the manuscript:
1. The rationale for the exclusion of patients with known cardiac disease should be discussed
2. The involvement of RV is novel and exciting. The author should include presentative images of the RV involvement: chamber dilation, RV volume, or any abnormalities.
3. The authors should include a discussion of current clinical management methods/treatment for post-covid cardiac complications
Author Response
Dear Editor Ms. Lillie Zhang,
Dear Reviewers,
We appreciate very much your careful review of our manuscript.
We have been able to resolve the issues raised by the reviewers, and have made appropriate changes accordingly.
Please find attached the detailed reply to all comments/suggestions.
Changes made in the manuscript are in red.
We trust that you will find these changes satisfactory and we look forward to hearing from you at your earliest convenience.
Yours sincerely,
on behalf of all authors
Marek Kozinski, MD, PhD
Reviewer No 2
Reviewer's comments:
Comments and Suggestions for Authors
The manuscript entitled “Cardiac Magnetic Resonance Findings in Patients Recovered 2 from COVID-19 Pneumonia and Presenting with Persistent 3 Cardiac Symptoms: the TRICITY-CMR Trial” sought to assess the prevalence of cardiac involvement in patients after COVID-19 recovery. The finding of RV involvement is exciting and novel. Below are some minor comments that can further strengthen the manuscript:
- The rationale for the exclusion of patients with known cardiac disease should be discussed
Reply: Thank you for this comment. In the revised Study Design and Population sub-section, we explained that: „We excluded from the study patients with known cardiac disease in order to minimize confounding (i.e., to reduce the probability of detecting non-COVID-19-related CMR abnormalities).”
- The involvement of RV is novel and exciting. The author should include presentative images of the RV involvement: chamber dilation, RV volume, or any abnormalities.
Reply: We are grateful for highlighting this finding of our study. The respective figure has now been included in the text.
Additionally, we wrote in the revised discussion:
“In our post-COVID-19 cohort, both RVEDV and the prevalence of reduced RVEF was significantly higher in hospitalized vs. non-hospitalized patients. Previous studies have demonstrated acute right ventricle dysfunction due to COVID-19 pneumonia and respiratory failure [32,33]. Moreover, COVID-19-related pulmonary thrombosis involving pulmonary microvessels may lead to an increase of pulmonary vascular resistance and probably increase mortality rates [34]. Paternoster at al. in their meta-analysis of 1,450 patients (half of them were invasively ventilated) have reported high mortality rates among patients with COVID-19 pneumonia requiring respiratory support and with right ventricular dysfunction, dilatation, or pulmonary hypertension [35]. Such abnormalities may potentially affect more likely patients requiring hospitalization with more severe respiratory symptoms.”
We also added to the Conclusions section: “Importantly, CMR imaging revealed lower RVEF and more frequent presence of reduced RVEF together with higher RVEDV in hospitalized vs. non-hospitalized post-COVID-19 patients.”
- The authors should include a discussion of current clinical management methods/treatment for post-covid cardiac complications
Reply: Thank you for this suggestion. We added to the revised manuscript a new discussion sub-section:
“4.4. Management of post-COVID-19 cardiac complications
A wide range of post-COVID-19 complications (e.g., myocarditis, myocardial infarction, right ventricular dysfunction/heart failure and arrhythmias) has been reported so far [36]. Their underlying pathomechanisms are still poorly understood. In general, the management of these complications does not differ from the management of corresponding non-post-COVID-19 cardiac diseases. A recent expert consensus on the management of patients recovered from COVID-19 pneumonia and presenting with cardiac symptoms recommends an initial diagnostic approach comprising of basic laboratory tests, including cardiac troponin, an electrocardiogram (ECG), an echocardiogram, ambulatory ECG Holter monitoring, chest imaging and/or pulmonary function tests (e.g., spirometry) [37]. Briefly, in patients with elevated cardiac troponin and/or ECG abnormalities indicating myocarditis and/or echocardiographic abnormalities, cardiology consultation is suggested. Furthermore, CMR examination is recommended in hemodynamically stable patients with suspected myocarditis. Importantly, antiviral treatment (e.g. therapy with remdesivir) should be restricted to patients with active SARS-CoV-2 infection.”

Round 2
Reviewer 1 Report
The Authors responded satisfactorily to all comments.